# Release of Bisphenol A from Pit and Fissure Sealants According to Different pH Conditions

**DOI:** 10.3390/polym14010037

**Published:** 2021-12-23

**Authors:** Eun-Deok Jo, Sang-Bae Lee, Chung-Min Kang, Kwang-Mahn Kim, Jae-Sung Kwon

**Affiliations:** 1Department and Research Institute of Dental Biomaterials and Bioengineering, College of Dentistry, Yonsei University, Seoul 03722, Korea; jdpink@hanmail.net (E.-D.J.); ridm@yuhs.ac (S.-B.L.); 2Department of Pediatric Dentistry, College of Dentistry, Yonsei University, Seoul 03722, Korea; kangcm@yuhs.ac; 3BK21 FOUR Project, College of Dentistry, Yonsei University, Seoul 03722, Korea

**Keywords:** pit and fissure sealant, dental biomaterials, pH, bisphenol A

## Abstract

Changes in intraoral pH can cause changes in the chemical decomposition and surface properties of treated resin-based pits and fissure sealants (sealant). The purpose of this study is to evaluate the release of bisphenol A (BPA) from sealants under three different pH conditions over time. The test specimen was applied with 6 sealants 5 mg each on a glass plate (10 × 10 mm) and photopolymerized. The samples were immersed for 10 min, 1 h, and 24 h in solutions of pH 3.0, 6.5, and 10.0 at 37 °C. BPA release was measured using a gas chromatography-mass spectrometer. A statistical analysis was performed by two-way ANOVA and one-way ANOVA to verify the effect of pH conditions and time on BPA release. The BPA concentration in the pH 3.0 group was higher at all points than with pH 6.5 and pH 10.0 (*p* < 0.05), and gradually increased over time (*p* < 0.05). As a result, it was confirmed that low pH negatively influences BPA release. Therefore, frequent exposure to low pH due to the consumption of various beverages after sealant treatment can negatively affect the sealant’s chemical stability in the oral cavity.

## 1. Introduction

Dental pits and fissure sealants (sealant) are among the most commonly used materials to prevent tooth decay in children and adolescents [1,2,3,4]. However, sealants consist mainly of bis-GMA, bis-EMA, and bis-EMA monomers containing bisphenol A (BPA, CAS number: 80-05-7) [5,6]. That is, BPA is a precursor to monomers, which are organic substrates. BPA may exist as an impurity if chemical synthesis is not completed in the process of preparing the dental sealant based on monomers such as bis-GMA, or if the synthesis reset does not reach stoichiometric completion. In 1996, Olea et al. [5] reported that leakage of BPA from dental sealants to patient saliva increased concerns about the potential estrogenicity of dental materials. Another study by Fung et al. [7] argued that BPA released from tooth sealants might be absorbed or present in undetectable amounts in the systemic circulation. BPA was recognized in the 1930s as an endocrine-disrupting chemical (EDC) that mimics estrogen and changes hormone function [8]. Since the 1990s, similar effects of BPA on female hormones have been reported [9,10,11]. In addition, the adverse effects of BPA on the human body include disorders in central nervous system development and function, reproductive function, thyroid hormone function, and fetal oocyte meiosis function [12,13]. In addition, BPA can be more harmful to children and adolescents, who are the main subjects of sealant treatments [14]. One study found that early exposure to BPA accelerates the onset of puberty in female mice but reduces reproductive parameters [13]. In another study, the concentration of BPA was 10 times higher than that of adults in neonatal rats when oral administration of biologically active BPA was detected [15]. Because of this, infants and toddlers are susceptible to BPA exposure, so the use of products containing BPA for infants and toddlers is completely prohibited in the United States, the United Kingdom, and Korea [16].

BPA is formed by the decomposition of bis-GMA or bis-EMA and is released in various concentrations depending on the chemical or mechanical processes occurring in the oral cavity [17]. Drinking various beverages causes changes in the pH of the saliva in the mouth [18]. Recently, the consumption of carbonated drinks, fruit juices, and sports drinks by children and adolescents has increased due to improved living standards and the spread of eating-out culture [19,20]. Beverages which are widely consumed have different pH levels. Reddy et al. [21] classified the pH of 379 beverages in one state of the USA, finding that 93% (354 of 379) were below pH 4.0. These acidic foods or beverages are essential factors that affect the durability and lifespan of resin recovery. Low acidity may cause deterioration of the physical properties and chemical deterioration of the restored material [22]. As a result of examining the difference in residual monomer leakage according to acidity and immersion time for three types of composite resins, it was reported that the outflow increased significantly as the immersion time increased at pH 4 [23].

Therefore, it is essential to consider the effect of pH levels on oral repair materials. However, there have been few studies on BPA release according to pH and time in dental sealants. In addition, the concentrations of BPA released from previously reported dental sealants are wide-ranging, making it difficult to draw comparisons. The reason for this is that the study method differs from the actual method of clinical sealant treatment and the amount of sealant used. Therefore, in this study, a research method was designed by investigating the amount of sealant used in clinical practice. The purpose of this study is to evaluate the release of bisphenol A (BPA) from sealants in 3 conditions of pH over time.

## 2. Materials and Methods

### 2.1. Materials

The sealants used in this study are shown in Table 1. Sealants used in this study were randomly selected from a resin-based light-cured type used commercially in clinical practice. The pH conditions of the solvent for immersing the specimens were classified into three levels: pH 3.0, pH 6.5, and pH 10.0. Step-by-step pH levels were measured and adjusted using a commercial pH meter (ORION™ Star A211, Thermo Scientific, Waltham, MA, USA). Before all procedures, the pH meter was calibrated using a pH buffer (Thermo Scientific™ Orion™ pH 4.01, 7.00, 10.01) for accurate reproduction. The pH 3.0 level was prepared by mixing lactic acid (CAS No. 50-21-5) in distilled water (JW-pharma. Co., Seoul, Korea), while the pH 10.0 level was prepared using sodium hydroxide solution (NaOH, CAS No. 1310-73-2). For the pH 6.5 level, sterile distilled water was used, which was opened immediately before the test and used after checking the pH. The prepared pH solutions were stored under air-tight conditions by packing them with a Press’n Seal^®^ (GLAD, Oakland, CA, USA).

### 2.2. Methods

#### 2.2.1. Pre-Investigation on the Amount of Sealant

The actual amount of sealant used in clinical practice in pit and fissure sealant treatments was investigated. Five dental hygienists with more than one year of clinical experience applied sealant to a tooth model, i.e., the second molar. The difference in mass before and after applying the sealant was analyzed. It was found that the average amount of total sealant was 4.90 mg (Table 2). Therefore, 5 mg of sealant was used in the BPA release test in this study.

#### 2.2.2. Sealant Specimens for BPA Release Test

5 mg of sealant was applied to the glass plate (10 mm × 10 mm). Sealant specimens were light-cured at the same distance, according to the manufacturer’s instructions. The total number of samples for the BPA release test was 270; five of each of the six sealants (Table 3).

#### 2.2.3. Procedure of BPA Release Test

First, the sealant (5 mg) was applied and photopolymerized to a glass plate (10 × 10 mm) to prepare each specimen. Next, the specimens were submerged completely in 2 mL solvent in a 15 mL conical tube. Next, these were immersed in an incubator shaker (Lab Companion SI-600, Seoul, Korea) at 37 °C for three time periods (10 min, 1 h, 24 h). Immediately after the immersion time, the specimens were removed from the conical tube, cooled in a freezer and then freeze-dried for more than 12 h using a freeze dryer (Ilshin Lab Co., Ltd., Yangju-si, Korea). BPA detection was analyzed by gas chromatography and a mass spectrometer (GC-MS; Agilent Technologies 7820A GC and 5977E MSD system, Palo Alto, CA, USA). Finally, 2 mL pure methanol (≥99.99%) was mixed in the conical tubes for GC-MS analysis. These were transferred to 1.5 mL vials (Agilent, Palo Alto, CA, USA) and used by GC-MS.

#### 2.2.4. Conditions and Calibration of GC/MS

The conditions of the GC-MS instrument for BPA detection are shown in Table 4. First, the molecular weight of BPA was confirmed by performing a qualitative analysis of standards (SCAN) according to the four BPA standard concentrations (10, 20, 50, 100 ppm). Then, to evaluate the amount of BPA detected for each sample, a quantitative analysis, i.e., selecting and measuring specific ions (selected ion monitoring, SIM), was performed. The standard material of BPA was prepared by mixing 1 g of bisphenol A (CAS: 80-05-7) and 1000 mg/L of methanol (≥99.99%) and then diluting with the same solvent to 10, 20, 50, and 100 ppm each. These solutions were stored at −18 °C. After GC-MS measurement of the samples, calibration was performed based on the standard concentration to obtain accurate BPA detection data. Figure 1 shows the calibration after GC-MS measurement of the standard samples. The resultant calibration functions had correlation coefficients (R^2^) ranging from 0.998 to 1.000.

### 2.3. Statistical Analysis

First, data on BPA release were separately subjected to a two-way ANOVA (pH level × immersion time) and Tukey’s test. Additionally, one-way ANOVA was performed to compare the difference in BPA concentration according to the pH group and time. Significance was determined at the *p* = 0.05 level.

## 3. Results

### 3.1. The Difference in BPA Concentration (ppm) According to pH Levels and Time

A comparison pf bisphenol A concentration (ppm) according to pH group and time is shown in Table 5. First, in the pH 3.0 group, the BPA concentration was higher at 24 h (2.14 ppm) than at 10 min (0.35 ppm) and 1 h (0.72 ppm) (F = 11.196, *p* < 0.05). Similarly, in the pH 6.5 and pH 10.0 groups, the BPA concentration was higher at 24 h (0.28, 0.52 ppm) than at 10 min (0.09, 0.09 ppm) or 1 h (0.18, 0.25 ppm) (F = 5.303, 9.189, *p* < 0.05). Next, regarding our comparison by time point, there were significant differences, i.e., the BPA concentration was higher at pH 3.0 than at pH 6.5 or pH 10.0 at all time points. At 10 min, the BPA concentration was higher at pH 3.0 (0.35 ppm) than pH 6.5 (0.09 ppm) or pH 10.0 (0.09 ppm) (F = 15.492, *p* < 0.05). Similarly, at 1 h and 24 h, the BPA concentration was higher at pH 3.0 (0.72, 2.14 ppm) than at pH 6.5 (0.18, 0.28 ppm) or pH 10.0 (0.25, 0.52 ppm) (F = 11.518, 13.158, *p* < 0.05).

Differences in BPA concentration (ppm) according to each factor are shown in Figure 2. First, as a result of comparing BPA emission according to pH levels (a), the pH 3.0 group was 5.7 times and 3.7 times higher than the pH 6.5 group and pH 10.0 group, respectively (*p* < 0.05). Next, as a result of comparing the release of BPA over time (b), 24 h was 5.5 times and 2.6 times higher than 10 min and 1 h, respectively (*p* < 0.05).

### 3.2. Comparison of BPA Concentration According to pH Level and Time of Each Sealant

Table 6 and Figure 3 show a comparison of BPA concentrations of each sealant product according to pH level and time. BPA was detected in all sealants under all conditions. In addition, in all sealants, the BPA concentration of the pH 3.0 group was higher than in the pH 6.5 and pH 10.0 groups (*p* < 0.05). Finally, the BPA concentrations of all pH groups increased over time (within 24 h; *p* < 0.05).

## 4. Discussion

Today, sealants are essential dental materials to prevent occlusal caries of teeth. However, there has been concern that BPA, an environmental hormone-disrupting substance, may be detected from these restorative materials. In addition, the beverages we consume have different pH levels, which alter the pH conditions in our mouths [18]. There has been concern that chemical and physical changes would occur to restorative materials if the oral pH changes after dental sealant treatment. Therefore, this study evaluated the difference in BPA release according to three pH conditions and immersion time after polymerized resin-based light-curing type dental sealant.

In this study, first, the amount of sealant used in the BPA release test was investigated. The reason for this was that the range of amounts of sealant applied in previous laboratory studies varied widely, and even the amount used in actual clinical practice was found to be different [5,24,25,26]. Therefore, five dental hygienists with more than one year of clinical experience applied five of the most widely used molar model sealants. As a result, it was confirmed that an average of 5 mg of sealant was used per tooth (*p* < 0.05).

The results of this study confirmed that there was a significant difference in BPA emission according to pH level and immersion time (*p* < 0.05). First, the detected BPA concentration of the pH 3.0 group was 5.7 times and 3.7 times higher than those of the pH 6.5 and pH 10.0 groups, respectively (*p* < 0.05). The pattern of these results was similarly confirmed in all six sealants. Similarly, several studies have reported that low pH beverages cause surface decomposition in resin composite materials [22,27]. In addition, in a survey of the effect of acidity on the chemical dissolution of composite resins, the outflow of monomers from a pH 4 solution was significantly increased compared to a pH 7 solution [23]. Sealants are continuously exposed to various types of accommodation environments in the mouth. Hydrolysis reaction by water and expansion of the matrix surface by water absorption are the main causes of the chemical decomposition of resin-based restorations [28]. Components eluted from most composite resins are non-China compound monomers, but even crosslinked resins can cause hydrolysis. Figure 4a shows the hydrolysis step of bis-GMA [29]. Hydrolysis occurs when the OC = O bond between the acyl group of resin molecules and oxygen is broken [30]. At this time, since pores are generated, a decomposition product appears, yielding bisphenol A dimethacrylate (BADGE), 2,2-bis[4(2,3-hydroxypropoxy)phenyl]propane (bis-HPPP), BPA, etc. (Figure 4b) [31,32]. Perhaps the reason for this is that BPA is relatively unsoluble in water, but generally dissolves well in acetic acid, benzene, ethanol, etc.; Log Kow (Octanol-Water Partition Coefficient) = 3.32. [33]. As such, acidity can accelerate the decomposition of the sealant of the resin substrate and destroy chemical stability.

The detected BPA concentration after 24 h were 5.5 times and 2.6 times higher than after 10 min and 1 h, respectively (*p* < 0.05). At pH 3.0, the BPA detection concentration was 0.35 ppm (10 min), 0.75 ppm (1 h), and 2.14 ppm (24 h), showing higher values over time. Similar to the effect of pH, the pattern of these results was confirmed in all six sealants. In other words, the longer acidic drinks stay in our mouths, the more BPA is detected over a 24-h period. In addition, BPA was detected at pH 6.5 and pH 10.0, but levels were lower than at pH 3.0, increasing in the order of 10 min (0.09 ppm, 0.09 ppm), 1 h (0.18 ppm, 0.24 ppm), and 24 h (0.28 ppm). After 24 h, the BPA concentrations (pH 0.3 = 2.14 ppm, pH 6.5 = 0.28 ppm, pH 10.0 = 0.52) were lower than those reported by Pulgar and colleagues (pH 1.0 = 6.5 ppm, Ph 7.0 = 7.8 ppm) [25]. Our results were nonetheless similar those reported by Arenholt-Bindslev and colleagues (0.3–2.8 ppm) [24]. In 2000, MANABE and colleagues reported that when a sealant (1 mg) was immersed in water for 24 h, BPA was detected at 0.02–0.09 ppm, i.e., lower than 0.28 ppm (pH 6.5, 24 h) under similar conditions. [34]. The reason for this discrepancy is likely the different research methods used 20 years ago, such as the amount of sealant used, sealant type, and GC-MS equipment type and function improvements.

Finally, BPA was detected from all silanes under all conditions. Since various BPA concentrations were detected for each sealant, the minimum–maximum range of BPA according to pH level in the total sealant was in the order of pH 3 (0.07~7.74 ppm), pH 10.0 (<0.01~2.81 ppm), and pH 6.5 (<0.01~1.06 ppm). The standard for BPA elution of plastic food containers is 0.05 ppm in the EU, 0.6 ppm in South Korea, and 2.5 ppm in Japan [35,36,37]. The use of BPA in the manufacture of some baby products, including baby bottles, has been prohibited in the EU and South Korea. In addition, the use of BPA in manufacturing raw materials for cosmetics in the EU and South Korea has been prohibited. Comparing the range of BPA concentrations by pH group in this study, there may be concerns about the stability of dental sealants which are harmful to the human body. It was impossible to compare and analyze the amount of BPA detected by sealant type in this study. Previous research on the solubility of synthetic resins relative to acidity and time found that this factor was influenced by the kind of resin monomer and the composition of filler [38]. Perhaps the reason for this is that hydrophilicity varies depending on the type of resin monomer. The sealant used in this study was randomly selected. The types and configurations of resin substrates for each sealant cannot be accurately classified due to manufacturer confidentiality. Considering these points, future studies should consider comparisons of BPA detection by resin monomer type, and the development of BPA substitutes with endocrine toxicity.

The limitation of this study is that it was undertaken only with simple immersion without considering the immersion ability that affects the decomposition of the resin-based restorative. However, as an in vitro research method, there were two significant findings: (1) The oral temperature conditions and research methods were chosen to resemble actual conditions; and (2) Only differences due to pH were evaluated by controlling other influencing factors in the laboratory. Therefore, in future studies, it is suggested that a complementary experiment be applied which takes into account the pH circulation rate of saliva in the oral cavity.

## 5. Conclusions

Within the limitations of this study, it was confirmed that low pH is a factor that negatively influences BPA release. Therefore, frequent exposure to low pH due to the consumption of various beverages after sealant treatment can negatively affect the sealant’s chemical stability in the oral cavity.

## Figures and Tables

**Figure 1 polymers-14-00037-f001:**
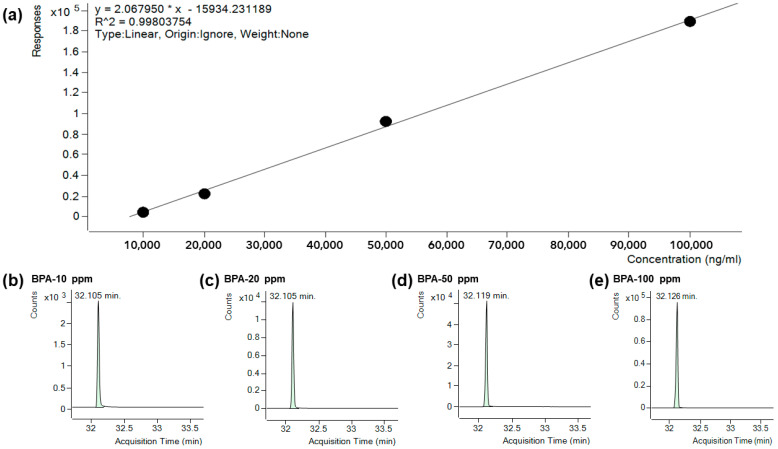
Calibration after GC-MS measurement of the standard samples: (**a**) Linear calibration of 4 samples, (**b**) Selected Ion (213.3) bisphenol A 10 ppm, (**c**) Selected Ion (213.3) bisphenol A 20 ppm, (**d**) Selected Ion (213.3) bisphenol A 50 ppm, (**e**) Selected Ion (213.3) bisphenol A 100 ppm.

**Figure 2 polymers-14-00037-f002:**
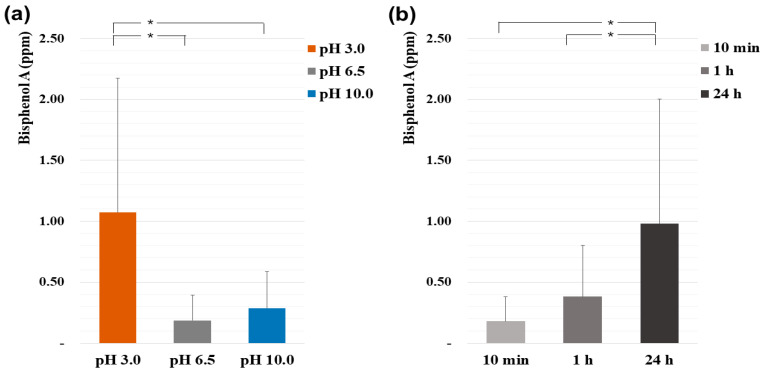
Differences in BPA concentration (ppm) according to each factor: (**a**) BPA release according to pH levels (* *p* < 0.05); (**b**) BPA release according to time (* *p* < 0.05).

**Figure 3 polymers-14-00037-f003:**
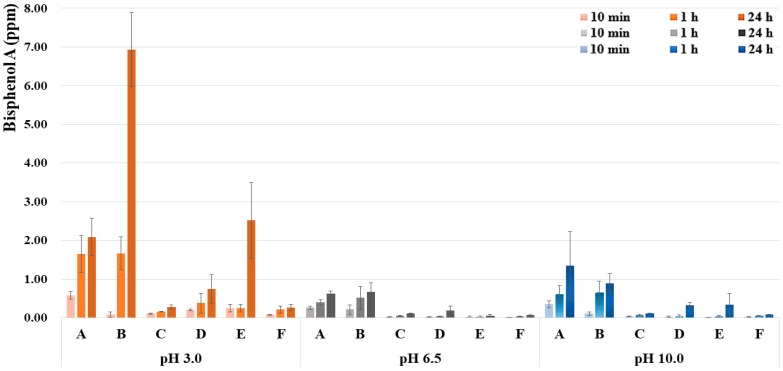
Comparison of BPA concentration (ppm) according to pH levels (*p* < 0.05) and time (*p* < 0.05) of each sealant (A: Clinpro^TM^, B: Eco-s^®^, C: UltraSeal XT^®^ plus, D: Charmseal^®^, E: Seal-it^®^, F: FORTIFY^®^).

**Figure 4 polymers-14-00037-f004:**
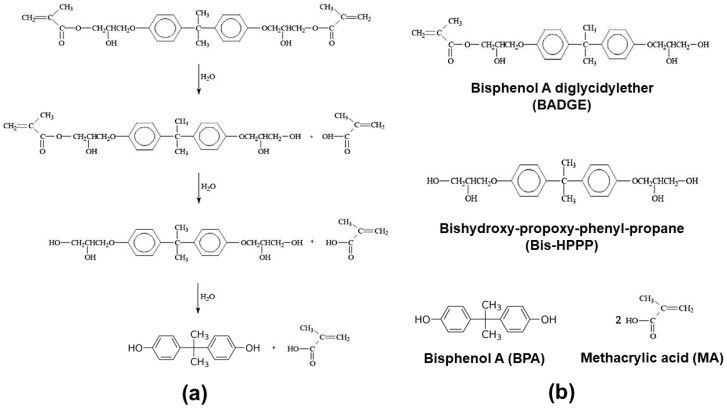
Hydrolysis of resin component: (**a**) the hydrolysis step of bis-GMA [29]; (**b**) the types of decomposition products.

**Table 1 polymers-14-00037-t001:** The compositions of the sealants were tested, according to the manufacturers’ information.

	Sealant	Composition (% by Wt)	Manufacturer
A	Clinpro^TM^	bis-GMA * (40~50), TEGDMA (40~50)	3M ESPE, Seefeld, Germany
B	Eco-s^®^	bis-GMA * (50~55), TEGDMA (35~40)	Vericom, Gyeonggi, Korea
C	UltraSeal XT^®^ plus	bis-GMA* (not revealed), TEGDMA (10~25), DUDMA (2.5~10)	Ultradent Products, South Jordan, UT, USA
D	Charmseal^®^	bis-GMA * (not revealed), TEGDMA, UDMA	DenKist, Gyeonggi, Korea
E	Seal-it^®^	bis-EMA * (30~50), TEGDMA (20~30)	Spident, Incheon, Korea
F	FORTIFY^®^	bis-DMA * (5~10), UDMA (30~50)	Bisco, Schaumburg, IL, USA

* BPA-based monomers, bis-GMA (bisphenol A glycidyldimethacrylate), TEGDMA (triethyleneglycol dimethacrylate), DUDMA (diurethane dimethacrylate), UDMA (urethane dimethacrylate), bis-EMA (bisphenol A ethoxylatedimethacrylate), bis-DMA (bisphenol A dimethacrylate).

**Table 2 polymers-14-00037-t002:** Amount of sealant applied to the first molar models.

No.	*N*	Amount of Sealant (mg)		F/*p*
		M ± SD	Min–Max	
1	5	4.47 ± 0.30	4.11–4.85	4.812/0.007
2	5	5.92 ± 0.70	4.69–6.47	
3	5	4.71 ± 0.71	4.06–6.01	
4	5	4.38 ± 0.39	4.00–4.98	
5	5	5.03 ± 0.70	4.07–5.77	
Total	25	4.90 ± 0.79	4.00–6.47	

**Table 3 polymers-14-00037-t003:** Sealant specimens for BPA release test.

Conditions	*N*	Sealant Specimens (ea)		
		10 min	1 h	24 h
pH 3.0	90	30 (6 S × 5 each)	30 (6 S × 5 each)	30 (6 S × 5 each)
pH 6.5	90	30 (6 S × 5 each)	30 (6 S × 5 each)	30 (6 S × 5 each)
pH 10.0	90	30 (6 S × 5 each)	30 (6 S × 5 each)	30 (6 S × 5 each)
Total	270	90	90	90

S (Sealants).

**Table 4 polymers-14-00037-t004:** Instrument conditions of GC-MS for BPA detection.

	Conditions			
Column	HP-5ms Ultra Inert (30 m 250 μm 0.25 μm)			
Oven Temp.	Unit	Rate (℃/min)	Temp. (℃)	Hold (min)
	Initial	-	40	0
	Ramp1	5	50	0
	Ramp2	5	80	2
	Ramp3	10	120	5
	Ramp4	10	280	1
	Ramp5	10	320	0
Inlet Temp.	250 ℃			
Injection Mode	splitless			
Injection Vol.	1 μL			
Carrier Gas	Helium			
Carrier Flow	0.7 mL/min			
Scan Parameters	40~615			
Sim Parameters	bisphenol A: 213.3, 119, 228, 214			

**Table 5 polymers-14-00037-t005:** The difference in BPA concentration (ppm) according to pH levels and time.

Group	BPA (ppm)				
	10 min M (SD)	1 h M (SD)	24 h M (SD)	(Min–Max)	F/*p*
pH 3.0	0.35 (0.30) ^*b*,*A*^	0.72 (0.73) ^*b*,*A*^	2.14 (2.55) ^*a*,*A*^	(0.07–7.74)	11.196 *
pH 6.5	0.09 (0.14) ^*b*,*B*^	0.18 (0.23) ^*ab*,*B*^	0.28 (0.28) ^*a*,*B*^	(<0.01–1.06)	5.303 *
pH 10.0	0.09 (0.13) ^*b*,*B*^	0.25 (0.31) ^*b*,*B*^	0.52 (0.60) ^*a*,*B*^	(<0.01–2.81)	9.189 *
F/*p*	15.492 *	11.518 *	13.158 *		
Source	F/*p*				
pH	24.440 *				
Time	18.153 *				
pH * Time	7.361 *				

^*a*,*b*^ Post-analysis by Tukey within a group (*a* > *b*), ^*A*,*B*^ Post-analysis by Tukey in a time point (A > *B*) * *p* < 0.05.

**Table 6 polymers-14-00037-t006:** BPA concentration (ppm) according to pH levels and time of each sealant.

Sealants/Group		Bisphenol A (ppm)			
		10 min M (SD)	1 h M (SD)	24 h M (SD)	F/*p*
A	pH 3.0	0.58 (0.10) ^*b*,*A*^	1.65 (0.47) ^*a*,*A*^	2.09 (0.48) ^*a*,*A*^	19.638 *
	pH 6.5	0.26 (0.05) ^*c*,*B*^	0.40 (0.07) ^*b*,*B*^	0.62 (0.08) ^*a*,*AB*^	35.813 *
	pH 10.0	0.35 (0.09) ^*b*,*B*^	0.61 (0.22) ^*ab*,*B*^	1.36 (0.87) ^*a*,*B*^	5.049 *
	F/*p*	19.457 *	24.396 *	8.192 *	
B	pH 3.0	0.09 (0.07) ^*b*,*A*^	1.67 (0.42) ^*b*,*A*^	6.93 (0.95) ^*a*,*A*^	145.406 *
	pH 6.5	0.22 (0.12) ^*b*,*B*^	0.52 (0.30) ^*ab*,*B*^	0.67 (0.24) ^*a*,*B*^	4.019 *
	pH 10.0	0.11 (0.05) ^*b*,*B*^	0.65 (0.30) ^*a*,*B*^	0.90 (0.24) ^*a*,*B*^	16.704 *
	F/*p*	33.311 *	16.905 *	186.455 *	
C	pH 3.0	0.11 (0.01) ^*b*,*A*^	0.16 (0.01) ^*b*,*A*^	0.28 (0.05) ^*a*,*A*^	42.419 *
	pH 6.5	0.03 (<01) ^*c*,*B*^	0.06 (<01) ^*b*,*B*^	0.11 (0.01) ^*a*,*B*^	201.652 *
	pH 10.0	0.04 (0.01) ^*c*,*B*^	0.07 (0.02) ^*b*,*B*^	0.12 (0.01) ^*a*,*B*^	64.039 *
	F/*p*	121.907 *	97.903 *	51.062 *	
D	pH 3.0	0.21 (0.02) ^*b*,*A*^	0.38 (0.25) ^*ab*,*A*^	0.75 (0.37) ^*a*,*A*^	5.780 *
	pH 6.5	0.03 (0.04) ^*b*,*B*^	0.04 (<01) ^*b*,*B*^	0.19 (0.12) ^*a*,*B*^	8.322 *
	pH 10.0	0.03 (0.01) ^*b*,*B*^	0.04 (0.04) ^*b*,*B*^	0.33 (0.06) ^*a*,*B*^	61.164 *
	F/*p*	374.387 *	8.902 *	8.138 *	
E	pH 3.0	0.25 (0.10) ^*b*,*A*^	0.25 (0.10) ^*b*,*A*^	2.52 (0.98) ^*a*,*A*^	4.188 *
	pH 6.5	0.02 (0.02) ^*B*^	0.02 (0.02) ^*B*^	0.05 (0.04) ^*AB*^	0.851
	pH 10.0	0.02 (<01) ^*B*^	0.04 (0.02) ^*B*^	0.34 (0.30) ^*B*^	2.462
	F/*p*	26.069 *	23.278 *	4.292 *	
F	pH 3.0	0.08 (0.01) ^*b*,*A*^	0.22 (0.09) ^*a*,*A*^	0.27 (0.08) ^*a*,*A*^	10.108 *
	pH 6.5	0.02 (<01) ^*c*,*B*^	0.04 (<01) ^*b*,*B*^	0.07 (<01) ^*a*,*B*^	781.970 *
	pH 10.0	0.02 (0.01) ^*c*,*B*^	0.05 (0.02) ^*b*,*B*^	0.09 (0.01) ^*a*,*B*^	49.230 *
	F/*p*	124.536 *	17.505 *	32.187 *	

A: Clinpro^TM^, B: Eco-s^®^, C: UltraSeal XT^®^ plus, D: Charmseal^®^, E: Seal-it^®^, F: FORTIFY^®^. ^*a*,*b*^ Post-analysis by Tukey within a group (*a* > *b*), ^*A*,*B*^ Post-analysis by Tukey in a time point (A > *B*), * *p* < 0.05.

## Data Availability

The data presented in this study are available on request from the corresponding author.

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
