# Peer review of "Release of Bisphenol A from Pit and Fissure Sealants According to Different pH Conditions"

_polymers, 2021, doi:10.3390/polym14010037_

Round 1

Reviewer 1 Report

The presented manuscript is addressed to the experimental study on the effect of the variation of pH on the release of Bisphenol A from pit and fissure sealants. The presentation is well organized but I have few recommendations:

  • Would you like to improve the quality of the figures? The letters are extremely small and it is very difficult for readers to analyze the information.
  • It is possible to present the data from the paragraph 3.2. in table or figure?

I recommend manuscript to be published in Polymers after major revision.

Author Response

Thank you very much for your kind reviews and comments regarding our manuscript (polymers-1503023) entitled above. Now we have carried out revisions according to your comments and hope this will be adequate for the acceptance of this manuscript. Details of corrections according to the comments are as follows:

Comment 1

Would you like to improve the quality of the figures? The letters are extremely small and it is very difficult for readers to analyze the information.

Response to Comment 1

Thank you for kind help on reviewing this article. Your valuable opinions have very assisted us to enhance the explanatory power of this paper.

According to your suggestion, we have now revised all the Figures. We adjusted the clarity of the Figures with adjustment of letter sizes.

Comment 2

It is possible to present the data from the paragraph 3.2. in table or figure?

Response to Comment 2

According to your suggestion, we now have presented the 'paragraph 3.2.' as new Table 6 and rewritten the result section of 3.2. as follows:

3.2. Comparison of BPA concentration according to pH levels and time of each sealant

‘Table 6’ and 'Figure 3' show the comparison of BPA concentrations according to pH levels and time of each sealant product. BPA was detected from all sealants under all conditions. In addition, in all sealants, the BPA of the pH 3.0 group was higher than the pH 6.5 and pH 10.0 groups (p<0.05). And BPA of all pH groups was gradually higher over time within 24 h (p<0.05).

Reviewer 2 Report

This work evaluated the bisphenol A (BPA) release from sealants after immersion in 3 pH conditions (pH 3.0, 6.5, and 10.0) over time. BPA release was measured by the Gas Chromatography-Mass Spectrometry. And BPA release varies depending on the composition of the sealants but was affected by acidity and time. It is well planned, the results are properly described and discussed, and the conclusions are sound and supported by the data. Thus, I recommend this manuscript for publication in polymers after some minor revisions towards the following points: 1.The authors should rephrase the abstract such that it better reflects the contents. 2.In Figure 4 (a) and (b), the reaction condition is H2O and acidity (pH 3-7) for the hydrolysis step of Bis-GMA, and the structure of Bis-GMA should be consistent. 3.In the introduction section, the references are old, and some closely related new literatures should also be cited in the manuscript.

Author Response

Thank you very much for your kind reviews and comments regarding our manuscript (polymers-1503023) entitled above. Now we have carried out revisions according to your comments and hope this will be adequate for the acceptance of this manuscript. Details of corrections according to the comments are as follows:

Comment 1

The authors should rephrase the abstract such that it better reflects the contents.

Response to Comment 1

Thank you for your thoughtful comments on the manuscript. According to your suggestion, we have rewritten the abstract to better reflect the content as follows:

Abstract: Changes in intraoral pH can cause changes in the chemical decomposition and surface properties of the treated resin-based pit and fissure sealants (sealant). The purpose of this study is to evaluate the release of bisphenol A (BPA) from sealants in 3 conditions of pH over time. The specimens for the test are applied with 6 sealants randomly selected 5 mg each on a glass plate (10  10 mm) and photopolymerized. The samples were immersed for 10 min, 1 h, and 24 h in solutions of pH 3.0, 6.5, and 10.0 at 37 °C. BPA release was measured using a gas chromatography-mass spectrometer. Statistical analysis was performed by two-way ANOVA and one-way ANOVA to verify the difference between pH and time on BPA release. The BPA concentration in the pH 3.0 group was higher at all points than pH 6.5 and pH 10.0 (p<0.05) and gradually increased over time (p<0.05). As a result, it was confirmed that low pH was a factor that negatively influenced BPA release. Therefore, frequent exposure to low pH due to consumption of various beverages after sealant treatment can negatively affect the sealant's chemical stability in the oral cavity.

Comment 2

In Figure 4 (a) and (b), the reaction condition is H2O and acidity (pH 3-7) for the hydrolysis step of Bis-GMA, and the structure of Bis-GMA should be consistent.

Response to Comment 2

Thank you for your kind review. According to your suggestion, Figure 4 has been modified to avoid confusion related to structure of each key chemicals.

Comment 3

In the introduction section, the references are old, and some closely related new literatures should also be cited in the manuscript.

Response to Comment 3

Sorry for lack of information. Accordance to your suggestion, we have now added closely related new references with some of newly published data as follows:

1. Introduction

Dental pit and fissure sealant (sealant) are among the most commonly used materials to prevent tooth decay in children and adolescents [1-4].

-          The references (2~4) added

2.          Ahovuo-Saloranta, A.; Forss, H.; Walsh, T.; Nordblad, A.; Mäkelä, M.; Worthington, H.V. Pit and fissure sealants for preventing dental decay in permanent teeth. Cochrane Db Syst Rev 2017, doi:https://doi.org/10.1002/14651858.cd001830.pub5.

3.          Azarpazhooh, A.; Main, P.A. Pit and fissure sealants in the prevention of dental caries in children and adolescents: a systematic review. Journal of the Canadian Dental Association 2008, 74, 171-178, doi:https://doi.org/10.1002/14651858.CD001830.pub5.

4.          Wiener, R.C.; Findley, P.A.; Shen, C.; Dwibedi, N.; Sambamoorthi, U. Acculturation and dental sealant use among US children. Community Dentistry and Oral Epidemiology 2021, doi:https://doi.org/10.1111/cdoe.12678.

Round 2

Reviewer 1 Report

I recommend its publications in Polymers.